# Neutrophil-Lymphocyte Ratio in Patients with Acute Heart Failure Predicts In-Hospital and Long-Term Mortality

**DOI:** 10.3390/jcm9020557

**Published:** 2020-02-18

**Authors:** Jun Hwan Cho, Hyun-Jai Cho, Hae-Young Lee, You-Jeong Ki, Eun-Seok Jeon, Kyung-Kuk Hwang, Shung Chull Chae, Sang Hong Baek, Seok-Min Kang, Dong-Ju Choi, Byung-Su Yoo, Kye Hun Kim, Jae-Joong Kim, Byung-Hee Oh

**Affiliations:** 1Heart Research Institute, Cardiovascular-Arrhythmia Center, College of Medicine, Chung-Ang University Hospital, Seoul 06973, Korea; cardio.jhcho@gmail.com; 2Department of Internal Medicine, Seoul National University Hospital, Seoul 03080, Korea; hylee612@snu.ac.kr (H.-Y.L.);; 3Department of Internal Medicine, Sungkyunkwan University College of Medicine, Seoul 06351, Korea; eunseok.jeon@samsung.com; 4Department of Internal Medicine, Chungbuk National University College of Medicine, Cheongju 28644, Korea; kyungkukhwang@gmail.com; 5Department of Internal Medicine, Kyungpook National University College of Medicine, Daegu 41944, Korea; scchae@knu.ac.kr; 6Department of Internal Medicine, The Catholic University of Korea, Seoul 06591, Korea; whitesh@catholic.ac.kr; 7Department of Internal Medicine, Yonsei University College of Medicine, Seoul 03722, Korea; smkang@yumc.yonsei.ac.kr; 8Department of Internal Medicine, Seoul National University Bundang Hospital, Seongnam 13620, Korea; djchoi@snu.ac.kr; 9Department of Internal Medicine, Yonsei University Wonju College of Medicine, Wonju 26426, Korea; yubs@yonsei.ac.kr; 10Heart Research Center of Chonnam National University, Gwangju 61469, Korea; KIJ10@lycos.co.kr; 11Department of Cardiology, Asan Medical Center, University of Ulsan College of Medicine, Seoul 05505, Korea; jjkim@amc.seoul.kr; 12Department of Internal Medicine, Seoul National University College of Medicine, Seoul 03080, Korea; ohbhmed@snu.ac.kr; 13Department of Cardiology, Mediplex Sejong Hospital, Incheon 21080, Korea

**Keywords:** acute heart failure, survival, mortality, outcome, neutrophil-lymphocyte ratio

## Abstract

The application of a simple blood test to predict prognosis in acute heart failure (AHF) patients is not well established. Neutrophil-lymphocyte ratio (NLR) is inexpensive and easy to obtain in hospitalized patients using a routine blood test. We evaluate the prognostic implications of NLR as an independent predictor of in-hospital and long-term mortality in AHF patients. Among 5625 patients enrolled in the Korean Acute Heart Failure registry, 5580 patients were classified into quartiles by their NLR level, and analyzed for in-hospital and post-discharge three-year mortality. Patients in the highest NLR quartile had the highest in-hospital and post-discharge three-year mortality. The same results were seen by dividing the aggravating factor into the infection or ischemia group and the non-infection or non-ischemia group. For patients aggravated from infection or ischemia, a cut-off NLR value was 7.0 that increase the risk of in-hospital and post-discharge three-year mortality. In subgroups of patients not aggravated from infection or ischemia, a cut-off NLR value was 5.0 that increase the risk of in-hospital and post discharge three-year mortality. Elevated NLR in AHF patients at the index hospitalization is an independent predictor for in-hospital and post-discharge three-year mortality. Taken together, NLR is a marker for risk assessment of AHF patients.

## 1. Introduction

Inflammation plays a crucial role in the pathogenesis and progression of cardiovascular disease. Numerous inflammatory biomarkers are correlated with disease severity and prognosis across throughout heart failure (HF) [1]. White blood cell (WBC) count and its subtypes are classical markers of inflammation in cardiovascular disease [2]. Leukocytosis increases incidence of HF hospitalization and mortality [3]. Moreover, neutrophilia is associated with increased incidence of acute decompensated heart failure in patients with acute myocardial infarction [4], and lymphopenia is related to poor prognosis in patients with HF [5,6].

Neutrophil-lymphocyte ratio (NLR), which neutrophils counts are divided by lymphocyte counts, is used as a new additional inflammatory markers [7]. Elevated NLR was recently reported to be an independent predictor of both all-cause mortality and cardiovascular disease [8,9]. Previous studies showed that NLR is related to increased mortality or heart transplantation risk in patients with chronic HF [10,11], but these studies are limited by single-center study design and relatively small sample size. Therefore, we aimed to evaluate the prognostic implications of NLR on in-hospital and post-discharge three-year mortality in patients with AHF.

## 2. Materials and Methods

### 2.1. Study Design and Data Collection

The KorAHF (Korean Acute Heart Failure) registry is a prospective, nationwide multicenter cohort that consecutively enrolled 5625 patients hospitalized for acute HF syndrome from 10 tertiary care hospitals throughout the country from March 2011 to February 2014. Detailed information on the study design and results or the KorAHF registry has been previously published elsewhere [12,13]. In brief, patients who had signs or symptoms of acute HF such as either pulmonary edema, objective findings of left ventricular systolic dysfunction, or structural heart disease were enrolled in the study and there were no other specific exclusion criteria. At index admission, information about patient demographics, medical history, symptoms and signs of HF, laboratory test results, electrocardiogram and echocardiography, current medication, hospital course, and clinical outcomes were collected. In February 2017, the three-years follow-up of all enrolled patients were completed. During the follow-up (at one month, three months, six months, one year, and annually thereafter), information about clinical outcomes, laboratory data, and medication has been collected. The study was conducted in accordance with the declaration of Helsinki, and written informed consent was obtained before enrollment. All study protocol were reviewed and approved by the Institutional Review Board of each participating hospital (H-1102-072-352) in February 2011 and also registered with ClinicalTrials.gov (NCT01389843).

### 2.2. Study Endpoints and Definitions

The main clinical outcomes of the study were adverse in-hospital outcomes (defined as all-cause mortality or urgent heart transplantation during hospitalization) and all-cause mortality during the minimum three-year follow-up after discharge. In-hospital mortality and cause of death were adjudicated by an independent event committee. Each category of death have been presented in previous publication [14]. This adjudication form was made with reference by the adjudication of death of typical 3-phase randomized control trials such as RELAX and RELAX-AHF-2 trial [15,16]. The mortality data for patients who were lost to follow-up were collected from the National Insurance database or National Death Records of Korea.

NLR was calculated that neutrophils counts are divided by lymphocyte counts, using the same blood samples drawn at admission. Participants were classified into quartiles by their NLR level.

Left ventricular ejection fraction (LVEF) values were measured using thansthoracic echocardiography performed during the index hospitalization. Quantitative calculation using the modified Simpson’s biplane method was recommended for LVEF measurement, but visually estimated LVEF, measured by echo-specialized cardiologists, was also accepted as valid for HF categorization. According to the 2016 European Society of Cardiology (ESC) HF guidelines, heart failure with preserved ejection fraction (HFpEF) was defined as LVEF ≥ 50% and heart failure with reduced ejection fraction (HFrEF) was defined as LVEF < 40%. Patients with LVEF between 40 and 49% were considered to have heart failure with midrange ejection fraction (HFmrEF) [17].

To evaluate the association between NLR quartiles and in-hospital and post-discharge three-year mortality by aggravating factor, the patients were divided into two groups according to the aggravating factor (patients whose aggravating factor was infection or ischemia, and patients whose aggravating factor was not infection or ischemia).

### 2.3. Statistical Analyses

Continuous variables are expressed as mean ± standard deviation and analyzed using the Student’ test, one-way ANOVA test or Kruskal–Wallis test. Categorical variables are presented as numbers and relative frequencies (percentages) and were compared with the Chi-square test. Cox proportional hazard regression model was used to identify the predictors associated with the NLR quartiles with in-hospital and post-discharge three-year mortality. Variables that were statistically significant in the univariate analysis were included in the multivariable model, except variables that had >10% missing values or that were closely related to other clinical variables. Post-discharge survival was assessed using Kaplan–Meier estimate, and the association between NLR quartiles and crude all-cause death was compared using Log-rank test. To find the cut-off value of the NLR, the multivariate restricted cubic splines methods was used. This method provide a useful tool for the analysis of the effect of a continuous predictor on an outcome when the relationship between an outcome (dependent) variable and the explanatory (indepednet) variables is not linear because this method allows for great flexibility in the form of the relationship between predictor and outcome [18,19]. All *p* values are two-sided, and a value of less than 0.05 was considered statistically significant. All data were analyzed using SPSS v25.0 (IBM Corporation, Armonk, NY, USA).

## 3. Results

Of the 5625 patients enrolled in the KorAHF registry, 45 patients were excluded because their total and differential WBC counts were not available (*n* = 24), or they were not identified whether they survived or died during the follow-up period (*n* = 21). The final study participants consisted of 5580 (99.2%) patients (Scheme 1). The baseline characteristics of the study population according to NLR quartile are listed in Table 1. Patients in the higher NLR quartile were older and had lower BMI and more history of hypertension, diabetes, ischemic heart disease, and cerebrovascular disease. These patients also demonstrated higher heart rates and increased incidence of New York Heart Association class III or IV. Laboratory findings presented that patients in the higher NLR quartile were associated with increased WBC counts, blood urea nitrogen, and serum creatinine, and decreased hemoglobin, albumin, and sodium levels. These patients had higher incidence of B-type natriuretic peptide (BNP) ≥ 500 pg/mL or NT-proBNP ≥ 1500 pg/mL.

### 3.1. In-Hospital Mortality and 3-Year Survival After Discharge

In-hospital mortality including urgent heart transplantation occurred in 331 (5.9%) of the cases during admission. Patients in the highest NLR quartile had a higher rate of in-hospital mortality (9.9%, *n* = 138) compared with those in any other three quartiles. Patients belonging to NLR quartile 4 had a longer duration of hospital stay and higher rate of intensive care unit/coronary care unit admission. The details of in-hospital clinical outcomes are shown in Appendix A.

Of the initial 5580 patients who were hospitalized with AHF, 5,312 patients (95.2%) survived to discharge. A total of 11 subjects were lost to follow-up, and the three-year follow-up rate after discharge was 99.8%. Overall, 1891 patients (35.7%) died during the three-year follow-up period. Figure 1 presents overall in-hospital mortality and three-year survival after discharge according to the NLR quartiles. As compared with quartile 1, overall in-hospital mortality and post-discharge three-year mortality were increasing in quartiles 2, 3, and 4 (*p* < 0.001, *p* < 0.001, respectively). Even at post-discharge one month and six months, the overall survival of four groups according to quartile showed a significant difference. In univariate and multivariate analyses and Cox regression analysis with NLR quartile 1 as the reference, the patients in NLR quartile 4 showed higher overall in-hospital mortality (adjusted odds ratio (OR) 2.23, 95% confidence interval (CI) 1.44–3.44) and post-discharge three-year mortality (adjusted OR 1.44, 95% CI 1.24–1.67) (Table 2, Appendix A).

The subgroup analysis included the evaluation of the association between NLR quartiles and in-hospital and post-discharge three-year mortality in patients classified according to the frequency of HF and to the severity of left ventricular systolic function (LVEF) (Appendix A). Regardless of the number of exacerbations of HF and the severity of LVEF, the higher the NLR, the worse the overall in-hospital survival and post-discharge three-year survival. In the patients of HFpEF, OR tends to increase from Quartile 3. In-hospital and post-discharge three-year mortality in Quartile 4 are 1.83 (95% CI, 0.60–5.56), 1.81 (95% CI, 1.34–2.42), respectively (Appendix A).

### 3.2. Relationship between NLR and Overall Survival According to the Aggravating Factor of HF

Subgroup analysis was performed by dividing the aggravating factor of HF into two groups [non-ischemia/non-infection group and ischemia/infection group] (Figure 2). In the non-ischemia/non-infection group, the baseline characteristics of the population according to NLR quartile are listed in Table 3. Patients in the higher NLR quartile were older and had lower BMI and more history of hypertension, diabetes, ischemic heart disease, and prior admission due to HF. Patients in the higher NLR quartile demonstrated higher heart rates and increased incidence of New York Heart Association class III or IV. Laboratory findings presented that patients in the higher NLR quartile were associated with increased WBC counts, blood urea nitrogen, and serum creatinine, and decreased hemoglobin, albumin, and sodium levels. These patients had higher incidence of BNP ≥ 500 pg/mL or NT-proBNP ≥ 1500 pg/mL and renal replacement therapy. In quartile 4, 65 patients (8.3%) died and showed the highest in-hospital mortality. As shown in Figure 1, the higher the NLR quartile, the lower the overall in-hospital survival and post-discharge three-year survival. In multivariate analysis, as compared with NLR quartile 1, the adjusted ORs of in-hospital mortality and post-discharge three-year mortality in quartile 4 were 2.39 (95% CI, 1.19–4.81) and 1.81 (95% CI, 1.40–2.35), respectively (Table 4A).

The trends of the baseline characteristics of the ischemia/infection group are similar to that of the non-ischemia/non-infection group (Table 5). Of the 613 patients in quartile 4, 69 patients (11.3%) died, with the highest rate among the 4 quartiles. Multivariate analysis showed that NLR quartile 4 showed a higher in-hospital mortality (adjusted OR 2.24, 95% CI 1.26–4.00) and post-discharge three-year mortality (adjusted OR 1.30, 95% CI 1.05–1.73) with NLR quartile 1 as the reference (Table 4B). The survival curves showed consistent results, as shown in Figure 2.

### 3.3. Cut-Off Value of NLR

The cut-off value of the overall in-hospital mortality and post-discharge three-year mortality was obtained using the spline method. When the aggravating factor was non-infection/non-ischemia, the appropriate NLR cut-off value of overall in-hospital mortality and post-discharge three-year mortality was 5; the adjusted ORs of overall in-hospital mortality and post-discharge three-year mortality were 2.12 (95% CI 1.33–3.37) and 1.61 (95% CI, 1.32–1.96), respectively (Table 6A). The in-hospital mortality and post-discharge three-year mortality curves were significantly different between NLR ≥ 5.0 and NLR < 5.0, as shown in Figure 3. However, when the aggravating factor was infection/ischemia, the appropriate NLR cut-off value of overall in-hospital mortality and post-discharge three-year mortality was 7; the adjusted ORs of in-hospital mortality and post-discharge three-year mortality were 1.91 (95% CI, 1.31–2.80) and 1.20 (95% CI, 1.03–1.47), respectively (Table 6B). The overall in-hospital mortality and post-discharge three-year mortality curves are shown in Figure 4.

## 4. Discussion

Our study shows important implication about the effect of NLR on in-hospital mortality and post-discharge mortality in patients with AHF. In AHF patients, higher NLR increases in-hospital mortality and mortality at one-month, six-month and three-year after discharge and NLR is an independent predictor for in-hospital and post-discharge three-year mortality.

In a previous study of approximately 1200 patients with acute decompensated heart failure followed up for 26 months, patients with higher NLR have significantly higher 30-day readmission and long-term mortality rate compared with those with lower NLR [10]. According to Benites-Zapata et al. [11], elevated NLR is related to increased mortality or heart transplantation risk in advanced HF. However, both studies are limited by small number of patients in single center and retrospective study.

Our study is a multi-center prospective study involving more than 5000 patients. In contrast to the two previous studies, we performed a subgroup analysis by the number of HF and the severity of LVEF according to the 2016 ESC guidelines. Regardless of the number of decompensated HF and the degree of LVEF, the higher the NLR, the worse the in-hospital mortality and the post-discharge three-year mortality. Particularly, as compared to the patients with HFrEF, the in-hospital and post-discharge three-year mortality were worse with increasing NLR in HFpEF patients. Considering the potential and emerging role of inflammation in HFpEF patients, NLR may be a good indicator for prognosis. In addition, a subgroup analysis was performed according to the aggravating factors of HF. In both infection/ischemia and non-infection/non-ischemia groups, in-hospital mortality and post-discharge three-year mortality were higher in the higher NLR group. As a result, NLR was independently associated with worse outcome.

No consensus has been reached on the cut-off values to define the levels of NLR as most studies classified into tertiles or quartiles. In our study, the cut-off value was 6, as calculated using the spline method. The cut-off value was different according to the aggravating factor of HF. When the aggravating factor was ischemia and/or infection, the cut-off value was 7 and the OR was 1.91 (95% CI, 1.31–2.80) in in-hospital mortality and 1.20 (95% CI, 1.03–1.47) in post-discharge three-year mortality. When the aggravating factor was not ischemia or infection, the cut-off value was 5 and the OR was 2.12 (95% CI, 1.33–3.37) in in-hospital mortality and 1.61 (95% CI, 1.32–1.96) in post-discharge three-year mortality.

As well known, BNP/NT_proBNP is an excellent marker for predicting prognosis in patients with heart failure [20,21]. However, there are a few problems for BNP, such as variability between kits and underestimation due to short-half life. NLR is inexpensive and has little variability during the test, compared to BNP. Furthermore, NLR can be usefully used in combination with BNP for various clinical settings.

We suggest two possible mechanisms why higher NLR increases all-cause mortality in patients with HF. First, NLR is an inflammatory marker. Second, NLR reflects sympathetic tone. NLR is a combination of two independent inflammatory markers involved in two different immune pathways: neutrophils as a marker of the ongoing nonspecific inflammation are involved with a much quicker response, and lymphocytes as a marker of the regulatory pathway are associated with more adaptive long-term response of the immune system, as called physiological stress [22,23,24]. When other different inflammatory stimulus occurs, Leukocytes in response release pro-inflammatory cytokines, such as acid phosphatase, elastase, and myeloperoxidase. The release of these cytokines has detrimental effects on the myocardium, which leads to decrease ventricular function [25,26,27]. By contrast, lymphocytes are related to the regulation of the immune system pathway [22]. Inflammation stimulate lymphocytopenia downregulation of the lymphocyte apoptosis, differentiation and lymphocyte proliferation, and neurohumoral activation [28,29]. Lymphopenia has been demonstrated to be an independent prognostic factor, being related to lower survival rates in patients with HF [6,30,31].

Another plausible mechanism is that NLR can be affected by autonomic nerve balance. In other words, a higher NLR could imply a higher ratio of sympathetic/parasympathetic tone. When sympathetic tone are stimulated, the number and function of granulocytes are increased. On the other hand, those of lymphocytes are increased by parasympathetic tone [32].

This study has several limitations. First, as the results were analyzed only with the NLR at the time of admission, evaluating the effect of changes in NLR values on clinical outcomes over time was not possible. Second, although multivariable adjustment was performed to clinical variable that are known to be significant, the possibility of potential confounders, such as nutritional status and concomitant inflammation caused by other comobidities, remains. Third, our study is a retrospective analysis of a prospective cohort, additional study might be needed. Fourth, adjustments to well-known factors of acute heart failure outcomes such as congestion may be lacking, and attention should be given to their interpretation.

## 5. Conclusions

NLR is a cost-effectiveness, easy to apply clinical applicable value that predicts short-term and three-year prognosis and stratifies the risk of patients with AHF. Elevated NLR in patients with AHF on admission is an independent predictor for in-hospital and post-discharge three-year mortality even if adjusted for well-known predictors and interactions. It is also independent predictor regardless of the severity of LVEF or aggravating factor of AHF. For practical and clinical applications, this study demonstrates the cut-off value of NLR to predict outcomes, instead of tertile or quartile values in the analyzed population. In general, the cut-off value of NLR is 6. When the exacerbation factor was infection or ischemia, the cut-off value of NLR is 7. However, when the exacerbation factor is not ischemia or infection, the cut-off value of NLR is 5. We suggest that our study results should be taken into consideration in treatment for patients with AHF in clinical settings.

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
