# Peer review of "Neutrophil-Lymphocyte Ratio in Patients with Acute Heart Failure Predicts In-Hospital and Long-Term Mortality"

_jcm, 2020, doi:10.3390/jcm9020557_

Round 1
Reviewer 1 Report
Cho et al. present a retrospective analysis of a prospective cohort study that assessed the prognostic implications of Neutrophil-lymphocyte ratio (NLR) as an independent predictor of short- and long-term mortality in 5,580 acute heart failure patients.
The relatively large study sample (based on multi-center prospective study) and the subgroup analysis (by the number of HF and the severity of LVEF according to the 2016 European Society of Cardiology guidelines) have the potential to provide important insight on the prognostic value of NLR in AHF patients.
The manuscript is well-written and data analysis is impressive. I believe a few comments still need to be addressed:
Major comments:
The association between NLR with Overall in –hospital mortality and 3-year survival after discharge needs to be assessed using Cox regression analysis (Figure 2, table 2). Based on table 1, the clinical characteristics of HF patients in the highest quartiles of NLR were clearly different (with a longer duration of hospital stay) these covariates potentially affect patient prognosis and therefore should be adjusted using the Cox regression. Please present the summary table including the adjusted odds ratio, 95% CI and p values) The sub-analysis presentation is too long. Perhaps you can join fig. 3 and fig. 4 into a single figure with two parts? You might want to consider removing the unadjusted OR to the supplementary material generating one 'easy to read' table. The cut off value to NLR; I am not familiar with the spline method. Could you elaborate why you decided to use this method (please add reference) instead of using more traditional methods (CHAID, Youden J-statistic). Also, please justify why you categorized all covariates to the selected cutoffs. Please revise the sentence "In addition, the role of elevated NLR in clinical outcomes in patients with acute heart failure (AHF) has not been well investigated" (page 2 line 63-64). According to your discussion a fair amount of research was done on NLR in AHF.Minor comments:
Revise the sentence "Follow-up of the registered patients was planned until the end of 2018 with regular visits" (methods, page 2 line 79) to indicate accurate 3 years end of follow-up visits. Change figure 1 to scheme 1- study design. Add fig. 5 &6 to the scheme. Add p-value to figure 5 & 6.Author Response
We thank the Reviewer for constructive criticism provided and for helpful comments which improved the presentation of our work. We have addressed all comments through the responses that follow the Reviewer’s comments. As recommended, we have revised the manuscript to ensure greater clarity. Revisions to the text are are indentified in underlined text in red font..
Comment 1: The association between NLR with Overall in –hospital mortality and 3-year survival after discharge needs to be assessed using Cox regression analysis (Figure 2, table 2). Based on table 1, the clinical characteristics of HF patients in the highest quartiles of NLR were clearly different (with a longer duration of hospital stay) these covariates potentially affect patient prognosis and therefore should be adjusted using the Cox regression. Please present the summary table including the adjusted odds ratio, 95% CI and p values) The sub-analysis presentation is too long. Perhaps you can join fig. 3 and fig. 4 into a single figure with two parts? You might want to consider removing the unadjusted OR to the supplementary material generating one 'easy to read' table.
▶ Response 1: Thank you for these suggestions. As per your recommendation, we have re-analyzed the association between the NLR and mortality rate using Cox regression analysis. Accordingly, we modified Figure 2 and Table 2 to present our new results. Covariates that were previously identified to affect outcomes of heart failure were selected. All figures in the main text have also been revised to reflect the change in analysis procedure.
As recommended, we also merged Figures 3 and 4 and present the revised version as Figure 2 A-B. To improve readability, we reformatted Tables 2 and 5 to include only the adjusted OR and p-value; other details, including the unadjusted OR, are described in Supplementary Tables 6 and 7.
Comment 2: The cut off value to NLR; I am not familiar with the spline method. Could you elaborate why you decided to use this method (please add reference) instead of using more traditional methods (CHAID, Youden J-statistic). Also, please justify why you categorized all covariates to the selected cutoffs.
▶ Response 2: Thank you for raising this issue. The relationship between an outcome (dependent) variable and the explanatory (independent) variable(s) is not linear. Restricted cubic splines provide a useful tool to analyze the effect of a continuous predictor on an outcome under these conditions, providing greater flexibility by not being depending on the form of the outcome variable. Cubic splines can be used in multiple linear regression, logistic regression and survival analysis – namely all techniques that are used to draw inferences about parameter estimates can be applied to the cubic spline polynomials. In our study, we used a restricted cubic spline model for analysis due to the non-linear relationship between the NLR and the mortality rate. This has been additionally described to the Methods, Statistical analysis section and reference papers have been added in the revised manuscript (page 5).
Comment 3: Please revise the sentence "In addition, the role of elevated NLR in clinical outcomes in patients with acute heart failure (AHF) has not been well investigated" (page 2 line 63-64). According to your discussion a fair amount of research was done on NLR in AHF..
▶ Response 3: We appreciate the kind comment. We have deleted this sentence because the context before and after was inaccurate, as you indicated.
Comment 4: Revise the sentence "Follow-up of the registered patients was planned until the end of 2018 with regular visits" (methods, page 2 line 79) to indicate accurate 3 years end of follow-up visits.
▶ Response 4: We agree with your comment. For clarity, we have deleted this sentence.
Comment 5: Change figure 1 to scheme 1- study design. Add fig. 5 &6 to the scheme. Add p-value to figure 5 & 6.
▶ Response 5: We agree with this suggestion and have changed Figure 1 to Scheme 1 and have added Figures 5 and 6 to the Scheme. As a result of these changes, our original Figure 5 is now Figure 3 and Figure 6 has been changed to Figure 4 (as the number of figures has decreased). We also added p-value to the revised Figures 3 and 4.

Reviewer 2 Report
I read with interest the work of Jun Hwan Cho et al on NLR ratio in acute heart failure patients. I congratulate the authors for the quality of the manuscript and of the analysis.
I have juste one main limitation that the authors should answer : BNP and NT pro BNP have been demonstrated for several years to be major predictive factors of rehospitalization and mortality in acute and chronic HF patients; indeed, in table 1, the 4th quartile of NLR ratio is associated with higher rates of increased BNP/NT proBNP.However, in the first multivariate analysis, BNP/NT proBNP levels are not taken into account to adjust the prognostic significance of NLR ratio.
Moreover, i may have miss something, but in the tables 3 and 4 the rates of patients with elevated BNP/NT proBNP don't seem consistent with those described in the whole population. And once again it is critical to adjust on BNP/NT proBNP levels the multivariate analysis in the subgroup analysis.
indeed, if the authors want to add a new biomarker in the setting of acute HF, they have to prove that NLR adds an incremental predictive value to BNP/NT proBNP, and this analysis is critical for the message carried in the paper. Otherwise, it is just another biomarker whose interest is questionable. So I suggest to add a univariate and multivariate analysis of the predictors of mortality in this population, including the NLR cut off that has been nicely described in this study but also BNP/NT proBNP cut off and to examine the interest of both biomarkers.
The comparaison of both prognostic values(NLR and BNP/NT proBNP) should also be discussed in the discussion chapter
Author Response
Comment 1: I have just one main limitation that the authors should answer: BNP and NT pro BNP have been demonstrated for several years to be major predictive factors of rehospitalization and mortality in acute and chronic HF patients; indeed, in table 1, the 4th quartile of NLR ratio is associated with higher rates of increased BNP/NT proBNP. However, in the first multivariate analysis, BNP/NT proBNP levels are not taken into account to adjust the prognostic significance of NLR ratio.
▶ Response 1: We completely agree with your comment. We have incorporated your suggestion throughout the manuscript. Multivariate analysis was performed again, including analysis of BNP/NT proBNP levels. The results of the analysis are described in revised Tables 2, 5 and 6.
Comment 2: Moreover, I may have miss something, but in the tables 3 and 4 the rates of patients with elevated BNP/NT proBNP don't seem consistent with those described in the whole population. And once again it is critical to adjust on BNP/NT proBNP levels the multivariate analysis in the subgroup analysis.
▶ Response 2: Thank you for your accurate and detailed comment. After careful review, we confirm that patients with elevated BNP/NT proBNP in Table 3 and Table 4 were listed incorrectly. We have revised these values in Tables 3 and 4.
Comment 3: Indeed, if the authors want to add a new biomarker in the setting of acute HF, they have to prove that NLR adds an incremental predictive value to BNP/NT proBNP, and this analysis is critical for the message carried in the paper. Otherwise, it is just another biomarker whose interest is questionable. So I suggest to add a univariate and multivariate analysis of the predictors of mortality in this population, including the NLR cut off that has been nicely described in this study but also BNP/NT proBNP cut off and to examine the interest of both biomarkers.
▶ Response 3: You have raised an important point. To address this issue, we stratified BNP/NT-proBNP into quartiles, as for the NLR, and compared the rate of in-hospital and post-discharge 3-year mortality between quartiles. Results were similar to those obtained for the NLR quartiles.
The cut-off value was calculated based on the NT-proBNP. The cut-off value was 9,541 pg/mL in the non-ischemia/infection group, yielding an adjusted OR of in-hospital and post discharge 3-year mortality of 2.04 (95% CI 1.21–3.46) and 1.79 (1.51–2.11), respectively. In the ischemia/infection group, the cut-off value was 3,636 pg/mL, yielding an adjusted OR of in-hospital and post discharge 3-year mortality of 2.18 (95% CI 1.34–3.49) and 1.74 (1.46–2.06), respectively. Detailed results of our analysis are shown in the Table and Figure below.
Additional Table for Reviewers. Univariate and multivariate logistic regression analyses for all-cause in-hospital and post discharge 3-year mortality among patients with HF, by subgroup analysis based on the presence/absence of aggravating factors and stratified by BNP/NT-proBNP cutoff value.
Patients whose aggravating factor were not infection or ischemia|
|
In-hospital mortality (n = 3,127) |
Post discharge 3 year mortality (n = 3,026) |
||
|
|
Unadjusted ORa |
Adjusted ORa |
Unadjusted ORb |
Adjusted ORb |
|
NT_proBNP < 9541 (pg/mL) |
1.0 (Ref.) |
1.0 (Ref.) |
1.0 (Ref.) |
1.0 (Ref.) |
|
NT_proBNP ≥ 9541 (pg/mL) |
2.81 (1.83–4.31) |
2.04 (1.21–3.46) |
2.31 (2.00–2.66) |
1.79 (1.51–2.11) |
Patients whose aggravating factor were infection or ischemia
|
|
In-hospital mortality (n = 2,453) |
Post discharge 3 year mortality (n = 2,286) |
||
|
|
Unadjusted OR |
Adjusted ORa |
Unadjusted OR |
Adjusted ORb |
|
NT_proBNP <3636 (pg/mL) |
1.0 (Ref.) |
1.0 (Ref.) |
1.0 (Ref.) |
1.0 (Ref.) |
|
NT_proBNP ≥3636 (pg/mL) |
1.93 (1.34–2.76) |
2.18 (1.34–3.49) |
2.16 (1.86–2.52) |
1.74 (1.46–2.06) |
Data are expressed as odds ratio (OR) and 95% confidence intervals (CI). Ref. = reference category.
aadjusted for age category (70> vs. 70≤ years), sex (male vs. female), body mass index category (25> vs. 23≤ kg/m2), etiology of heart failure (ischemic vs. non-ischemic), systolic blood pressure (100> vs. 100≤ mmHG), history of hypertension, history of diabetes mellitus, history of cerebrovascular disease, history of chronic obstructive disease, prior admission history of HF, presence of tachyarrhythmia on admission, sodium level (135> vs. 135≤ mmil/L ), creatinine level (2.0> vs. 2.0≤ ), left ventricular ejection fraction (40%> vs. 40%≤),
badjusted for age category (70> vs. 70≤), sex (male vs. female), body mass index category (25> vs. 23≤), etiology of heart failure (ischemic vs. non-ischemic), systolic blood pressure (100> vs. 100≤), history of hypertension, history of diabetes mellitus, history of cerebrovascular disease, history of chronic obstructive disease, prior admission history due to HF, presented tachyarrhythmia on admission, sodium level (135> vs. 135≤), creatinine level (2.0> vs. 2.0≤ mmil/L), left ventricular ejection fraction (40%> vs. 40%≤), AA, BB, RASi
The incremental value of NLR was also assessed by developing a sequence of Cox models, starting with multivariable factors, adding the cut-off value of NT_proBNP and finally the cut-off value of NLR. In the non-ischemia/infection group, the global chi-square of the statistical model increased from 509.3 to 536.4 (P<0.001). In the ischemia/infection group, the global chi-square of the statistical model increased from 378.7 to 382.5 (P=0.042). In conclusion, our results show that the NLR, like the BNP/NT_proBNP, is a contributive marker for predicting the prognosis of patients with AHF. Also the combination of NLR and BNP/NT-proBNP significantly improves to predict the prognosis of patients with AHF.
Comment 4: The comparison of both prognostic values (NLR and BNP/NT proBNP) should also be discussed in the discussion chapter.
▶ Response 4: We agree with this comment and have included a discussion of the prognostic values of both the NLR and the BNP/NT proBNP in our revised revised manuscript (page 10), as follows:
The BNP/NT-proBNP level has previously been reported as an excellent prognostic marker among patients with HF [20,21]. However, there are limitations to using the BNP, including variability between kits and the possibility of underestimation due to short-half life. By comparison, the NLR is inexpensive and exhibits little variability during the test. Furthermore, the NLR could be useful when its use is combined with the BNP in various clinical settings.
van Veldhuisen, D. J.; Linssen, G. C.; Jaarsma, T.; van Gilst, W. H.; Hoes, A. W.; Tijssen, J. G.; Paulus, W. J.; Voors, A. A.; Hillege, H. L., B-type natriuretic peptide and prognosis in heart failure patients with preserved and reduced ejection fraction. J Am Coll Cardiol 2013, 61 (14), 1498-506. Daniels, L. B.; Maisel, A. S., Natriuretic peptides. J Am Coll Cardiol 2007, 50 (25), 2357-68.

Round 2
Reviewer 1 Report
Thank you for your comments and clarifications. I have no further comments. Good luck!
Author Response
.
Reviewer 2 Report
I congratulate the authors for the new analysis they performed in such a short period of time. These new results add strong evidence on the hypothesis raised by the authors.
Author Response
.